# Weather Radar Parameter Estimation Based on Frequency Domain Processing: Technical Details and Performance Evaluation

**Shuai Zhang [1], Yubao Chen [1,\*], Zhifeng Shu [2], Haifeng Yu [3], Hui Wang [4], Jianjun Chen [1] and Lu Li [1]**

1   Radar Meteorological Centre, China Meteorological Administration, Beijing 100081, China; shuaizhang@cma.gov.cn (S.Z.)
2   School of Atmospheric Physics, Nanjing University of Information Science & Technology, Nanjing 210044, China
3   Huayun Metstar Radar Co., Ltd., Beijing 100085, China
4   Beijing Meteorological Observation Centre, Beijing 100081, China
\*   Correspondence: chenyb@cma.gov.cn

**Abstract:** Parameter estimation is important in weather radar signal processing. Time-domain processing (TDP) and frequency-domain processing (FDP) are two basic parameter estimation methods used in the weather radar field. TDP is widely used in operational weather radars because of its high efficiency and robustness; however, it must be assumed that the received signal has a symmetrical or Gaussian power spectrum, which limits its performance. FDP does not require assumptions about its power spectrum model and has a seamless connection to spectrum analysis; however, its application performance has not been fully validated to ensure its robustness in an operational environment. In this study, we introduce several technical details in FDP, including window function selection, aliasing correction, and noise correction. Additionally, we evaluate the performance of FDP and compare the performance of FDP and TDP based on simulated and measured weather in-phase/quadrature (I/Q) data. The results show that FDP has potential for operational applications; however, further improvements are required, e.g., windowing processing for signals mixed with severe clutter.

**Keywords:** weather radar; parameter estimation; frequency domain processing; simulation

## 1. Introduction

Weather radar is an indispensable active remote sensing observation equipment in the meteorological field and plays an important role in precipitation estimation [1,2], hydrometeor classification [3,4], and microphysical retrieval [5,6]. The principle of weather radar can be summarized in the following three steps (as shown in Figure 1): (1) weather radar emits electromagnetic waves into the atmosphere; (2) when the electromagnetic waves "touch" targets (e.g., rain, snow, hail, and other non-meteorological targets that are not the focus of this study) along their propagation path, scattering occurs in all directions and a back-scattering signal is received by the radar; (3) valuable information about the scattering targets (e.g., their size, phase, shape, and orientation) can be extracted by properly processing the received signal. This third step is key to determining whether the radar can provide any operational benefits.

The received signal usually undergoes processes such as amplification, mixing, filtering, and digitization [7] and finally generates in-phase/quadrature (I/Q) data (also known as time-series data or Level I data) represented by a series of complex voltages. The I/Q data are the sum of the scattering signals of all randomly distributed targets in the sampling volume and can be approximated as a Gaussian random process [8]. For meteorological researchers, operational personnel, and other users of weather radar, I/Q data can be difficult (and unnecessary) to understand. The base data (including several radar variables

such as the reflectivity factor at horizontal polarization ($Z_H$), radial velocity ($v_r$), spectrum width ($\sigma_v$), differential reflectivity ($Z_{DR}$), differential phase ($\phi_{DP}$), and co-polar correlation coefficient ($\rho_{HV}$)) and their patterns—that intuitively reflect atmospheric dynamics and microphysical characteristics—are more in line with their actual requirements. The process of "translating" I/Q data into base data is called parameter estimation in the field of weather radar and is an important link in weather radar signal processing [9].

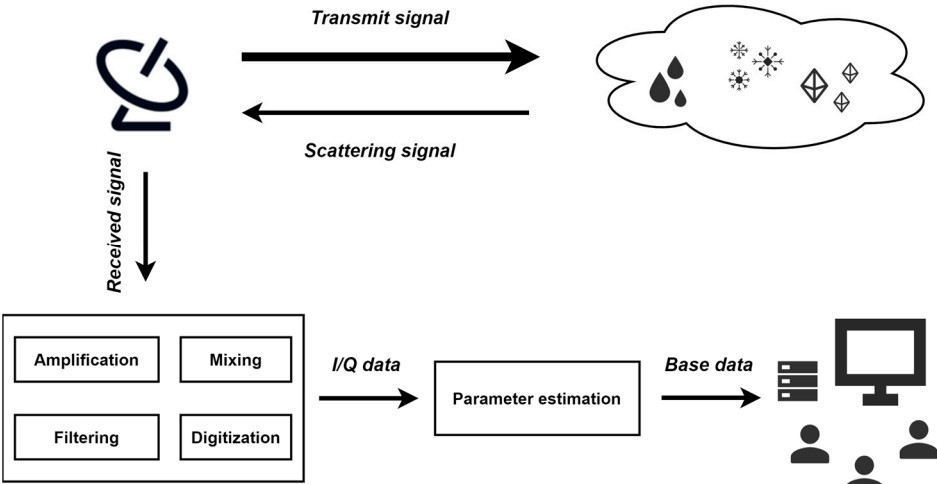

**Figure 1.** Schematic diagram of the weather radar principle presented in the form of signal/data flow.

There are two basic parameter estimation approaches—namely, frequency domain processing (FDP) using the power/cross-spectrum and time-domain processing (TDP) using the auto/cross-correlation function. Each approach has advantages and disadvantages; however, the essential information available is identical [9]. Because weather radar signal processing has extremely high requirements for real-time performance, TDP has been more widely used in operational weather radar systems than FDP owing to its advantages in terms of operating efficiency. Owing to the wide operational application of TDP, its problems have been exposed as quickly and comprehensively as possible. Through decades of investigations by researchers and engineers in the field of weather radar, existing TDP problems have been continuously explored, analyzed, optimized, and solved.

Melnikov and Zrnic [10] proposed one-lag TDP for dual-polarization weather radars, which used 1-lag correlations to estimate $Z_{DR}$ and $\rho_{HV}$, making them immune to noise. This estimator assumed that $\sigma_v$ of the horizontal and vertical polarization was consistent, which was expected to hold for most precipitation observed at elevation angles lower than 20°. However, the standard deviation (SD) of one-lag TDP was larger than that of conventional TDP (based on zero-lag) when $\sigma_v$ was broadening (e.g., wider than 6 m/s). Lei et al. [11] proposed a multi-lag TDP to improve parameter estimation performance at a low signal-to-noise ratio ($SNR$). The principle was to estimate the zero-lag correlations (affected by noise) by performing Gaussian fitting on the non-zero-lag correlations (immune to noise). The evaluation results showed that the bias and/or SD of estimated radar variables could be significantly improved if $\sigma_v$ was low or moderate. However, this estimator had greater requirements for sample number ($M$) and Nyquist velocity ($v_a$) [12]. To reduce the impact of noise, conventional TDP directly subtracts the noise power from zero-lag correlations. Noise power is typically measured after each volume scan. Specifically, the blue-sky noise power is routinely measured as part of an online system calibration performed at a high antenna elevation angle, which is scaled for use at lower antenna elevations using predetermined correction factors to account for thermal radiation from the ground [13]. Because noise may vary considerably at different azimuth angles, the noise power estimation mentioned above may not be representative of a specific location, which can cause substantial bias in the estimation of polarimetric variables, especially $\rho_{HV}$.

Ivic et al. [12] proposed a radial-based noise power estimation technique that improved the temporal and spatial resolutions of the noise power and thus reduced the bias of $\rho_{HV}$ estimates. Ivic [14,15] later proposed a series of improved and hybrid TDP methods to further reduce the number of invalid values of $\rho_{HV}$.

Currently, TDP still has inherent defects that have no solutions. For example, pulse-pair processing is the most common method for estimating $\overline{v}$ and $\sigma_v$ in TDP [16]. However, the estimation of $v_r$ ($\sigma_v$) assumes that the received signal has a symmetrical (Gaussian) power spectrum. Janssen and Speck [17] performed a statistical analysis of the power spectrum of weather signals, and their results showed that the non-Gaussian (e.g., asymmetrical or multipeaked) power spectrum accounted for approximately one-fourth of the total amount. Research results in [18–21] showed that the power spectra from some severe convective weather signals (e.g., tornados, hail, and lightning) presented bimodal or broad-spectrum characteristics. When the non-Gaussian weather signals appear, the TDP assumptions are no longer satisfied, and there will be an estimation bias if TDP is still used.

Compared to TDP, FDP exhibits inferior operating efficiency. However, with the development of computer technology, FDP can satisfy real-time operational requirements [22]. The most important advantage of FDP is its seamless connection to spectral analysis, which makes it more flexible for clutter suppression [23,24], range and Doppler ambiguity resolution [25,26], and severe weather identification [27,28]. Additionally, FDP does not require assumptions to be made regarding the power spectrum model; therefore, it can be used to estimate radar variables accurately, even if there is a non-Gaussian power spectrum signal. However, the application performance of FDP must be fully validated to ensure its robustness in an operational environment. This paper introduces some of the technical details of FDP. Moreover, in this study, the performance of FDP and a comparison between FDP and TDP were evaluated using ideal weather I/Q data generated based on the signal simulation technique and measured weather I/Q data.

The remainder of this paper is organized as follows: Section 2 describes the signal simulation technique, FDP, and conventional TDP (using TDP instead for the sake of brevity). Section 3 introduces several details of FDP (including window function selection, aliasing correction, and noise correction) and analyzes their performance based on simulated weather I/Q data. Section 4 presents a comparison of the performance of FDP and TDP based on simulated and measured weather I/Q data. Finally, a summary and conclusions are presented in Section 5.

## 2. Simulation and Estimation Methods

### 2.1. I/Q Data Simulator

The most well-known single-polarization I/Q data simulator in the weather radar field is the one proposed by Zrnic [29], which has served us well in developing and testing new weather radar signal processing algorithms over the past few decades [10,24]. Curtis [30] introduced several modifications to its accuracy and performance while keeping the basic framework unchanged; this was adopted in this study. To simulate the dual-polarization I/Q data, two realizations of single-polarization I/Q data were combined using the approach proposed by Galati and Pavan [31]. All the simulations mentioned in this paper are for weather signals and use the Gaussian spectral model.

The simulator inputs are $v_a$, $M$, the echo power for horizontal polarization ($P_h$), $v_r$, $\sigma_v$, $Z_{DR}$, $\phi_{DP}$, $\rho_{HV}$, and the noise power from the horizontal and vertical channels ($N_h$ and $N_v$). The simulator outputs are the I/Q data for the horizontal and vertical polarization channels ($V_h(m)$ and $V_v(m)$; argument $m$ denotes the $m$th pulse, which is a non-negative integer less than $M$). The simulation process can be summarized as follows:

1. Set the simulation length $M_s = \max{(2k+1, M+k)}$, where $k = ceil\left[\frac{v_a}{\pi\sigma_v}\sqrt{\frac{\ln 10}{5}A_T}\right]$. The auto-correlation threshold ($A_T$) can be calculated as follows:

$$A_T = \min\left\{25,\ 10 + \frac{5}{\ln 10}\left[\frac{\pi\sigma_v(M-1)}{v_a}\right]^2\right\}; \tag{1}$$

2.  Generate an ideal Gaussian power spectrum ($S_h(f)$; the argument $f$ denotes the spectral index) based on the given mean (i.e., $v_r$) and SD (i.e., $\sigma_v$), as follows:

$$S_h(f) = \frac{1}{\sqrt{2\pi\sigma_v^2}} e^{-(v(f)-v_r)^2/2\sigma_v^2}, \tag{2}$$

which can be computed on an extended Nyquist co-interval (from $-lv_a$ to $lv_a$), where $l$ denotes an integer factor, and its setting depends on the spectral threshold ($S_T = P_h - N_h + 35$). Specifically, the spectrum must be extended to cover the frequency range up to $S_T$ below its peak. Then, alias the extended spectrum to produce a spectrum on the desired Nyquist co-interval (from $-v_a$ to $v_a$);

3.  Set all values in the spectrum to zero, which is greater than $S_T$ below the peak of the spectrum;

4.  Appropriately scale the $S_h(f)$ so that the signal power is equal to the desired $P_h$;

5.  Simulate an independent and identically distributed complex Gaussian random process $W(f)$ with zero mean, unit variance, and $M_s$ length;

6.  Multiply $W(f)$ by the square root of the result of Step 4 and perform the inverse discrete Fourier transform (IDFT) to transform from the frequency domain to the time domain, as follows:

$$V_h(m) = \frac{1}{M_s}\sum_{f=0}^{M_s-1}\sqrt{S_h(f)}W(f)e^{j2\pi mf/M_s}; \tag{3}$$

7.  Repeat Steps 2–6 to generate $V_{h2}(m)$ (the extra "2" in the subscript is for the convenience of distinguishing it from $V_h(m)$). $V_v(m)$ can be calculated as follows:

$$V_v(m) = [\rho_{HV}V_h(m) + \sqrt{1-\rho_{HV}^2}V_{h2}(m)]\frac{e^{j\phi_{DP}}}{\sqrt{Z_{DR}}}; \tag{4}$$

8.  Return the first $M$ samples from the $M_s$ simulated samples;

9.  To add noise to the simulated signal, an independent and identically distributed Gaussian random process $W_h(m)$ ($W_v(m)$) is generated with zero mean, variance $N_h$ ($N_v$), and $M$ length. Then, add it to $V_h(m)$ ($V_v(m)$).

### 2.2. TDP

The auto-correlation function from the horizontal or vertical polarization ($\hat{R}_{h,\,v}(n)$) and the cross-correlation function ($\hat{C}_{hv}(n)$) estimated from $V_h(m)$ and $V_v(m)$ can be expressed as follows [8]:

$$\hat{R}_{h,v}(n) = \frac{1}{M-n}\sum_{m=0}^{M-n-1}V_{h,v}^*(m+n)V_{h,v}(m), \tag{5}$$

$$\hat{C}_{hv}(n) = \frac{1}{M-n}\sum_{m=0}^{M-n-1}V_h^*(m+n)V_v(m), \tag{6}$$

where $n$ denotes the lag number and $\wedge$ denotes the estimated value.

Once the correlation functions are obtained, the radar variables can be estimated using TDP. The specific estimation equations are listed in the first column of Table 1 [8]. $\hat{Z}_H$ can be estimated from $\hat{P}_h$ [22], as follows:

$$Z_H = 10\log_{10}\hat{P}_h + c + 20\log_{10}s + As, \tag{7}$$

where $c$ and $s$ denote the radar constant and slant range from the radar, respectively, $A$ denotes the two-way gaseous attenuation correction factor, which is sensitive to radar bands, and the default values of the S, C, and X bands are 0.016, 0.019, and 0.024 dB/km, respectively.

**Table 1.** Estimation equations for TDP and FDP.

| TDP | FDP |
|---|---|
| $\hat{P}_{h,v} = \hat{R}_{h,v}(0) - N_{h,v}$ | $\hat{P}_{h,v} = \sum\limits_{f=0}^{M-1} \hat{S}_{h,v}(f)$ |
| $\hat{v}_r = \angle \hat{R}_h(1)\frac{\lambda}{4\pi T_s}$ | $\hat{v}_r = \frac{\sum_{f=0}^{M-1} v(f)\hat{S}_h(f)}{\hat{P}_h}$ |
| $\hat{\sigma}_v = \sqrt{2\ln\frac{\lvert\hat{P}_h\rvert}{\lvert\hat{R}_h(1)\rvert}}\frac{\lambda}{4\pi T_s}$ | $\hat{\sigma}_v = \sqrt{\frac{\sum_{f=0}^{M-1}(v(f)-\hat{v}_r)^2\hat{S}_h(f)}{\hat{P}_h}}$ |
| $\hat{Z}_{DR} = 10\log_{10}\frac{\hat{P}_h}{\hat{P}_v}$ | $\hat{Z}_{DR} = 10\log_{10}\frac{\hat{P}_h}{\hat{P}_v}$ |
| $\hat{\phi}_{DP} = \frac{180}{\pi}\angle\hat{C}_{hv}(0)$ | $\hat{\phi}_{DP} = \frac{180}{\pi}\angle\sum\limits_{f=0}^{M-1}\hat{S}_{hv}(f)$ |
| $\hat{\rho}_{HV} = \frac{\lvert\hat{C}_{hv}(0)\rvert}{\sqrt{\hat{P}_h\hat{P}_v}}$ | $\hat{\rho}_{HV} = \frac{\lvert\sum_{f=0}^{M-1}\hat{S}_{hv}(f)\rvert}{\sqrt{\hat{P}_h\hat{P}_v}}$ |

*2.3. FDP*

The first step of FDP is to perform the discrete Fourier transform (DFT) on $V_{h,\,v}(m)$ to generate the complex amplitude spectrum ($F_{h,\,v}(f)$) [32], as follows:

$$F_{h,v}(f) = \frac{1}{M}\sum_{m=0}^{M-1}d(m)V_{h,v}(m)e^{-j2\pi mf/M}, \tag{8}$$

where $d(m)$ denotes the data window.

The second step is to estimate $\hat{S}_{h,\,v}(f)$ and the cross-spectrum ($\hat{S}_{hv}(f)$) as follows [32]:

$$\hat{S}_{h,v}(f) = \lvert F_{h,v}(f)\rvert^2 - \frac{N_{h,v}}{M}, \tag{9}$$

$$\hat{S}_{hv}(f) = F_h(f)F_v^*(f). \tag{10}$$

Once $\hat{S}_{h,v}(f)$ and $\hat{S}_{hv}(f)$ are obtained, the radar variables can be estimated using FDP. The specific estimation equations are listed in the second column of Table 1 [32,33]. The $\hat{Z}_H$ estimation of FDP is the same as that of TDP.

## 3. FDP Details

### 3.1. Window Function Selection

The I/Q data are obtained by finite pulse sampling; thus, it can cause spectral leakage owing to the discontinuity of the ends when performing DFT—that is, the energy at a specific frequency can spread to other frequencies [8]. In general, a window function is used to mitigate spectral leakage, which is a set of coefficients with the same length as the I/Q data and has a maximum value centered on it, tapering to near zero at the ends [34]. However, windowing can produce side effects such as reducing the number of effective samples and power loss. This study provides a quantitative assessment of these effects based on simulations. To ensure the universality of our conclusions, we designed two sets of simulation experiments (Sim1 and Sim2) for two typical precipitation types, namely, convective and stratiform precipitation.

The parameters used in the simulations are listed in Table 2. For simplicity, we set $N_{h,v}$ to 0 dB so that $P_{h,v}$ can be regarded as $SNR_{h,v}$. Considering that the echo intensity of convective precipitation is higher than that of stratiform precipitation, we set the $P_h$ of convective precipitation to a large value (30 dB), whereas that of stratiform precipitation was set to a relatively small value (15 dB), but not to a value that is significantly affected by noise. Based on a statistical analysis of the actual observations and a theoretical analysis of the physical model [35,36], $\sigma_v$, $Z_{DR}$, and $\rho_{HV}$ of stratiform precipitation were set at 1.5, 0.5, and 0.99, respectively, while those of convective precipitation were set at 3.5, 2.5, and 0.98, respectively. Because the value of $v_r$ ($\phi_{DP}$) has no effect on the simulation results, $v_r$ ($\phi_{DP}$) for both types of precipitation was set at 0 m/s (50°). For each experiment,

10,000 realizations of I/Q data were simulated with a $v_a$ of 26.8 m/s—that is, a pulse repetition frequency of 1000 Hz and radar frequency of 2.8 GHz. $M$ was set at 16, 32, 64, 128, and 256.

**Table 2.** Parameters used in the simulation. $N_{h,v}$ is set at 0 dB, $v_a$ is set at 26.8 m/s, and $r$ is set at 10,000 in all simulations.

| | $P_h$ (dB) | $v_r$ (m/s) | $\sigma_v$ (m/s) | $Z_{DR}$(dB) | $\phi_{DP}$ (deg) | $\rho_{HV}$ | $M$ |
|---|---|---|---|---|---|---|---|
| Sim1 | 30 | | 3.5 | 2.5 | | 0.98 | 16, 32, 64, 128, 256 |
| Sim2 | 15 | | 1.5 | 0.5 | 50 | 0.99 | |
| Sim3 | | 0 | 1, 2, 4 | | | | 64 |
| Sim4 | 30 | | 4 | / | / | / | 16, 32, 64 |
| Sim5 | | 16.8, 21.8, 23.8, 25.8 | 2.5 | | | | 64 |
| Sim6 | 0, 5, 10, 15, 20, 25, 30 | | | 1.5 | 50 | 0.985 | |
| Sim7 | 30 | 0 | 0.5, 1, 1.5, 2, 2.5, 3, 3.5, 4, 4.5 | | | | 128 |
| Sim8 | 0, 5, 10, 15, 20, 25, 30 | | | / | / | / | |
| Sim9 | 30 | | 2.5 | | | | 16, 32, 64, 128, 256 |
| Sim10 | 30, 25, 20 | −12, 0, 12 | | | | | |
| Sim11 | 30 | −10, 10 | | | | | 64 |

The parameters used in the FDP are listed in Table 3. In addition to using a rectangular window (i.e., without using a window function) when performing DFT, the window functions commonly used in weather radar signal processing were used to compare their performance, including the Hamming, Hann, Chebyshev (50 dB), Blackman, and Nuttall windows [23,24,33]. Noise correction was not used as the $SNR$ was sufficiently high such that the additional impact of noise correction was not introduced.

**Table 3.** Parameters used in the FDP.

| | Window Function | Aliasing Correction | Noise Correction |
|---|---|---|---|
| Sim1 | Rectangle, Hamming, Chebyshev, Hann, Blackman, Nuttall | | |
| Sim2 | | | |
| Sim3 | Hamming | No | No |
| Sim4 | | | |
| Sim5 | | CS, CP | |
| Sim6 | | | ZT, HY |
| Sim7 | Rectangle, Hamming (only for $\sigma_v$) | | |
| Sim8 | | CP | |
| Sim9 | | | HY |
| Sim10 | | | |
| Sim11 | | | |

The bias and SD of the radar variables can be used to quantify the simulation results for an objective evaluation. Using $P_h$ as an example:

$$Bias(P_h) = \frac{1}{r}\sum_{i=1}^{r}\hat{P}_h(i) - P_h, \tag{11}$$

$$SD(P_h) = \sqrt{\frac{1}{r}\sum_{i=1}^{r}\left[\hat{P}_h(i) - P_h\right]^2}, \tag{12}$$

where $r$ denotes the number of realizations, $\hat{P}_h(i)$ denotes the estimation result of the $i$th simulation, and $P_h$ denotes the simulation input, which can be considered to be the true value.

The bias in the estimation of the radar variables for convective precipitation is shown in Figure 2. $P_h$ estimates of the other window functions—except for the rectangular window—cause a significant power loss. Although there is some alleviation as $M$ increases, the bias still exceeds 4 dB for 256 samples (typically not achievable by operational weather radar systems). Other window functions have a smaller $v_r$ estimate bias compared to that of the rectangular window when $M$ is small, the maximum difference being only approximately 0.15 m/s at 16 samples. For the $\sigma_v$ estimates, all other window functions have a smaller bias than that of the rectangular window, indicating that spectral leakage has been effectively alleviated. Although the difference between the two decreases as $M$ increases, it is still close to 1 m/s for 256 samples. The bias of the estimation results of polarimetric variables (i.e., $Z_{DR}$, $\phi_{DP}$, and $\rho_{HV}$) is essentially the same under the different window functions.

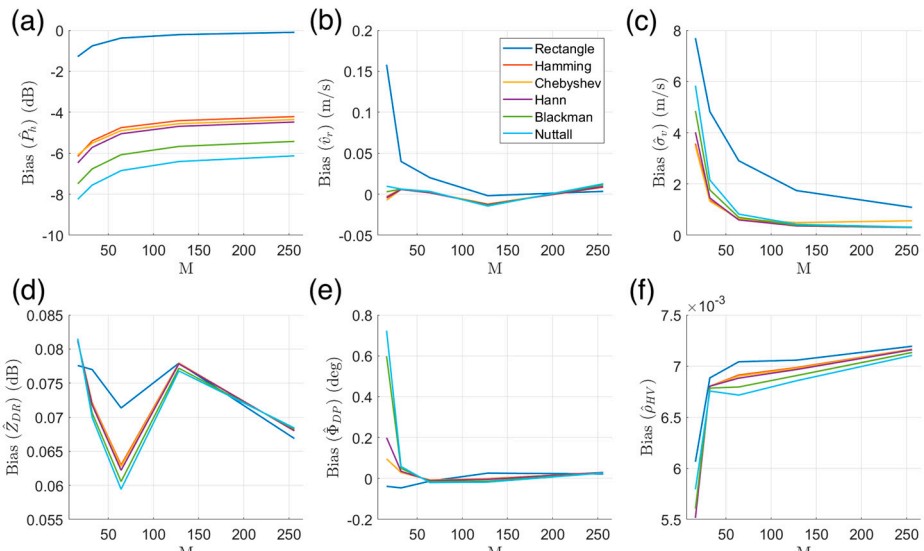

**Figure 2.** Bias of FDP for convective precipitation for different $M$ and window functions. (**a**) $\hat{P}_h$; (**b**) $\hat{v}_r$; (**c**) $\hat{\sigma}_v$; (**d**) $\hat{Z}_{DR}$; (**e**) $\hat{\phi}_{DP}$; and (**f**) $\hat{\rho}_{HV}$.

The SDs of the radar variable estimations for convective precipitation are shown in Figure 3 and clearly show that there are two situations. First, for radar variables other than $\sigma_v$, the estimation results using the rectangular window have an SD smaller than those of other window functions, and the SD increases as the taper of the window function increases. This can be understood as the effect of reducing the number of effective samples. Conversely, the $\sigma_v$ estimates have a maximum SD when using the rectangular window, whereas the SD differences between the other window functions are negligible.

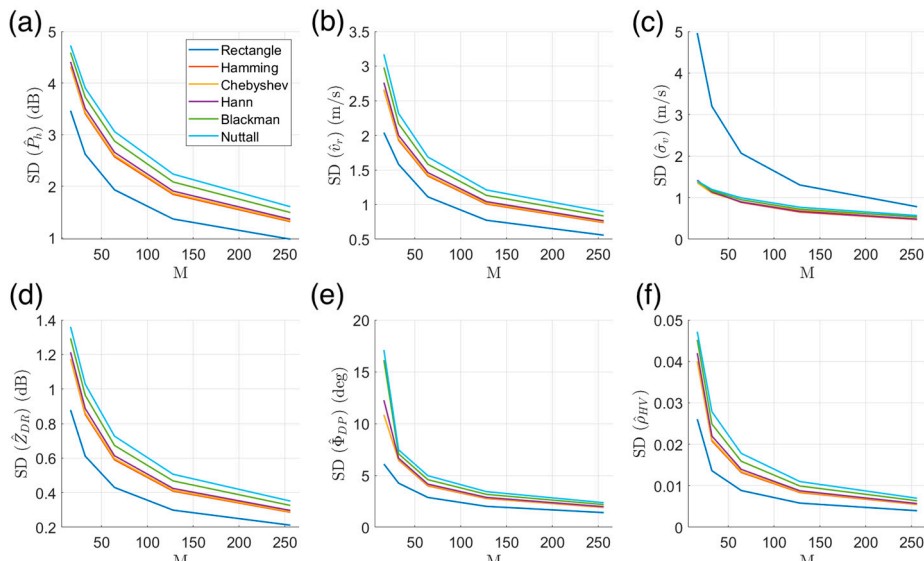

**Figure 3.** SD of FDP for convective precipitation for different $M$ and window functions. (**a**) $\hat{P}_h$; (**b**) $\hat{v}_r$; (**c**) $\hat{\sigma}_v$; (**d**) $\hat{Z}_{DR}$; (**e**) $\hat{\phi}_{DP}$; and (**f**) $\hat{\rho}_{HV}$.

The bias and SD of the radar variable estimations for stratiform precipitation are shown in Figures 4 and 5; the most significant difference between them and those of convective precipitation lies in the $\sigma_v$ estimates. Compared with Figure 2c, the bias of the $\sigma_v$ estimates using the rectangular window is less significant compared to those using other window functions (Figure 4c). Unlike Figure 3c, the $\sigma_v$ estimate using the rectangular window has the lowest SD (Figure 5c). Additionally, the $\sigma_v$ estimates of stratiform precipitation have a characteristic that its bias and SD decrease as the taper of the window function decreases (except for the rectangular window).

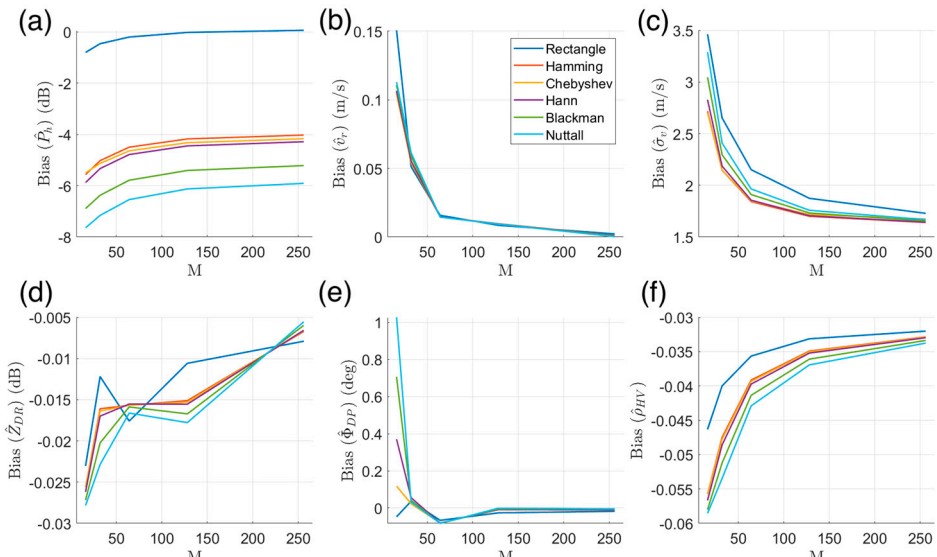

**Figure 4.** Bias of FDP for stratiform precipitation for different $M$ and window functions. (**a**) $\hat{P}_h$, (**b**) $\hat{v}_r$, (**c**) $\hat{\sigma}_v$, (**d**) $\hat{Z}_{DR}$, (**e**) $\hat{\phi}_{DP}$, and (**f**) $\hat{\rho}_{HV}$.

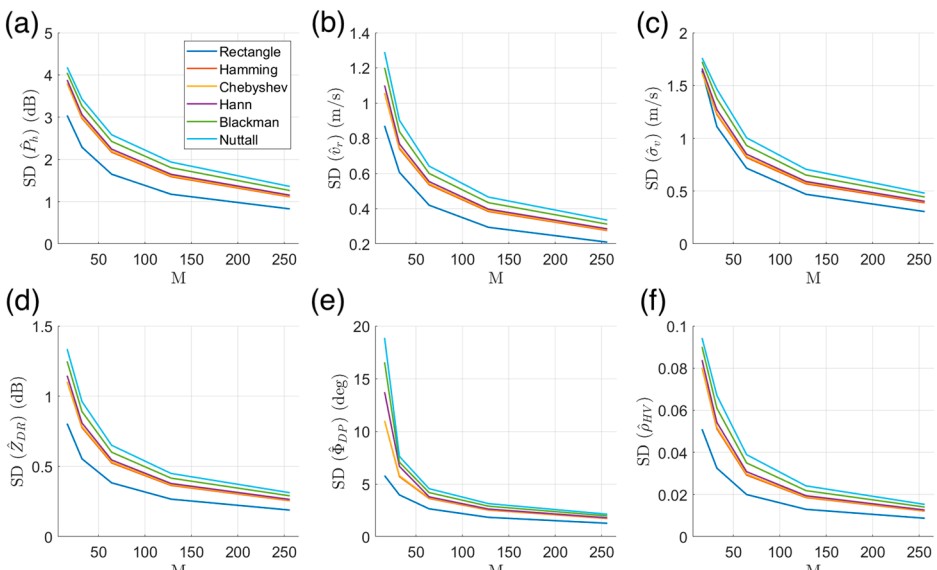

**Figure 5.** SD of FDP for stratiform precipitation for different $M$ and window functions. (**a**) $\hat{P}_h$, (**b**) $\hat{v}_r$, (**c**) $\hat{\sigma}_v$, (**d**) $\hat{Z}_{DR}$, (**e**) $\hat{\phi}_{DP}$, and (**f**) $\hat{\rho}_{HV}$.

It should be noted that the power loss caused by the window functions can be compensated for by normalizing the window function. The normalized window function $d_N(m)$ can be calculated as follows [24,32]:

$$d_N(m) = \frac{d(m)}{\sqrt{\frac{1}{M}\sum_{m=0}^{M-1}d(m)^2}}. \tag{13}$$

The performance of this method was evaluated based on simulations (Sim3 and Sim4), which generated known power I/Q data for different $\sigma_v$ (1, 2, and 4 m/s) and $M$ (16, 32, and 64). $P_h$ ($SNR_h$) was set to a large value (30 dB) to avoid noise. Because only a power analysis was required, a single-polarization I/Q data simulation was performed (polarimetric variables were not set). Consideration of the other input parameters was similar to that in the simulation mentioned above. Only the Hamming window was used as an example in the analysis. The difference between the $P_h$ estimates using the normalized Hamming window and those using the rectangular window (regarded as its true value)—that is, $\Delta P_h$—is shown in Figure 6 using a histogram. The mean and SD of $\Delta P_h$ with different $\sigma_v$ and $M$ are listed in Table 4. It is evident that as $\sigma_v$ or $M$ decreases, the distribution becomes less concentrated (an increase in SD as shown in Table 4), gradually sloping toward the positive side (an increase in the mean as shown in Table 4). Consequently, the adoption of this method decreases the quality of the power estimates. Therefore, even if this method is used for power compensation, it results in a decrease in the performance of $P_h$ estimates.

Based on the above analysis, we can conclude that the use of window functions (except for the rectangular window) in performing DFT is beneficial for improving the accuracy of $\sigma_v$ estimates. For other radar variables, the use of window functions (except for the rectangular window) results in a decrease in the number of effective samples and an increase in the SD. Therefore, two types of DFT are performed in FDP—that is, one for $\sigma_v$ estimates using a window function with a low taper (the default is the Hamming window), and another for other radar variable estimates using a rectangular window.

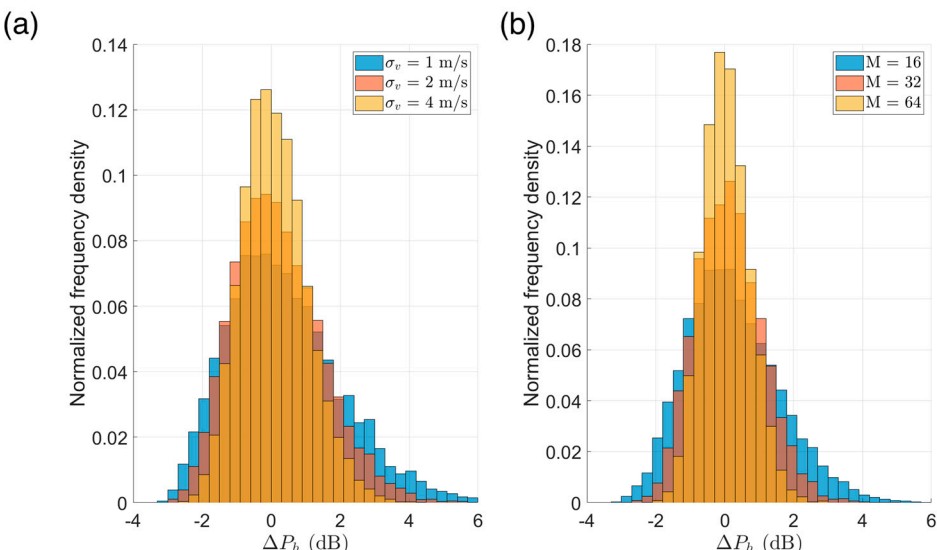

**Figure 6.** Difference between the $P_h$ estimates using the normalized Hamming window and the $P_h$ estimates using the rectangular window. (**a**) For different $\sigma_v$; (**b**) For different $M$.

**Table 4.** Mean and SD of the difference between the $P_h$ estimates using the normalized Hamming window and the $P_h$ estimates using the rectangular window with different $\sigma_v$ and $M$.

| | $\sigma_v$ | | | $M$ | | |
|---|---|---|---|---|---|---|
| | **1 m/s** | **2 m/s** | **4 m/s** | **16** | **32** | **64** |
| Mean | 0.374 dB | 0.194 dB | 0.101 dB | 0.223 dB | 0.121 dB | 0.04 dB |
| SD | 1.641 dB | 1.282 dB | 0.948 dB | 1.344 dB | 0.96 dB | 0.677 dB |

### 3.2. Aliasing Correction

Figure 7a shows a schematic of the power spectrum of a typical weather signal. The change in $v_r$ of the target detected by the radar is represented by the translation of the entire distribution along the horizontal axis. In the field of radar meteorology, the positive (negative) $v_r$ is specified as moving away from (toward) the radar, which corresponds to the movement in the direction indicated by the red (green) arrow in Figure 7a.

Based on the sampling theorem, $v_a$ is the maximum measurable $v_r$ of the radar (i.e., $-v_a$ to $v_a$ is the measurement range of $v_r$) [8]. As the absolute value of $v_r$ increases, a part of the power spectrum exceeds this measurement range. At this time, a spectrum-aliasing phenomenon appears—that is, the part of the power spectrum that exceeds this measurement range moves back into the measurement range from the other side (Figure 7b). As shown in the second row, second column of Table 1, the physical meaning of the $v_r$ estimates of FDP is the power-weighted average spectral Doppler velocity. Consequently, when spectrum aliasing occurs, it inevitably causes a bias in the $v_r$ estimates, as well as a bias in the $\sigma_v$ estimates (the third row, second column of Table 1). The $v_r$ and $\sigma_v$ estimates are collectively referred to as Doppler estimates in the following discussion.

This paper introduces two spectrum-aliasing correction methods. The first one is named the circular shifting method (CS), and its procedure is described as follows:

1. Find the spectral Doppler velocity with maximum power in the power spectrum ($v_{max}$), which can be used as an approximate $v_r$ estimate.
2. The power spectrum distribution can be rearranged by circular shifting, such that the corresponding position of $v_{max}$ is adjusted to zero. This can cause the estimation results to be immune to or minimized by spectrum aliasing.
3. Perform the Doppler estimates of FDP using the equations given in Table 1.
4. Add $v_{max}$ to the estimation result of $v_r$ in Step 3 to obtain the final $v_r$ estimates.

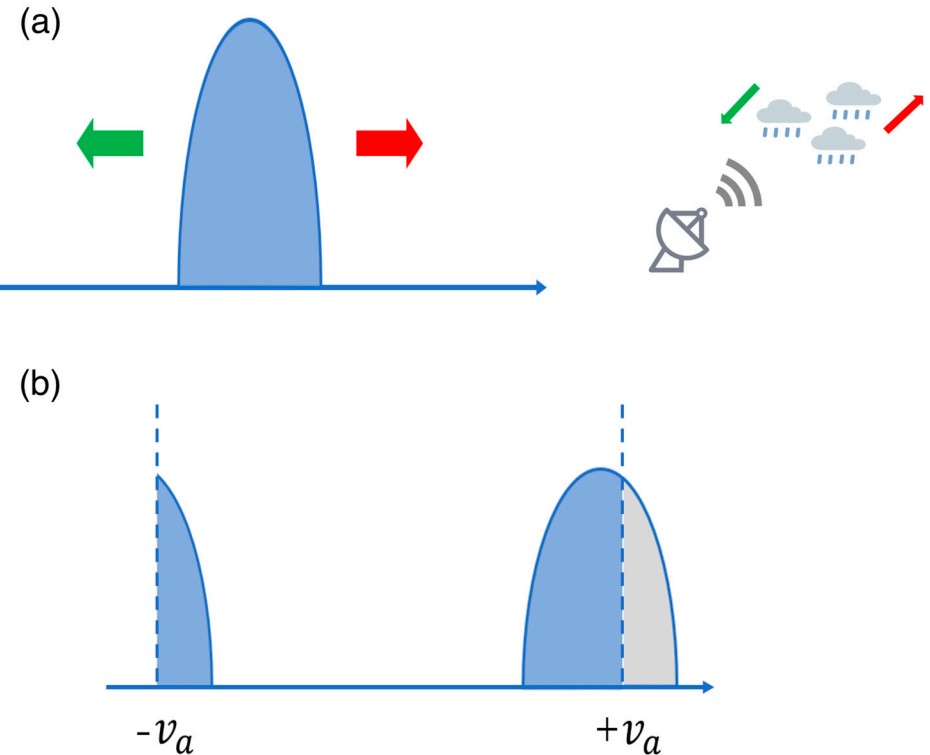

**Figure 7.** Schematic diagram of the power spectrum for a typical weather signal. (**a**) Response to the motion of weather targets; (**b**) Distribution splitting during spectrum aliasing.

The second method is the complex plane method (CP). Its principle is to represent the velocity as a complex number and perform Doppler estimates in the complex plane to avoid the discontinuity caused by the real-valued velocity. The modified Doppler estimation equations are [33]:

$$\hat{v}_r = \frac{v_a}{\pi} \angle [\sum_{f=0}^{M-1} \hat{S}_h(f) e^{j\pi v(f)/v_a}], \tag{14}$$

$$\hat{\sigma}_v = \sqrt{\frac{v_a^2}{\pi^2 \hat{P}_h} \sum_{f=0}^{M-1} \hat{S}_h(f) [\angle e^{j\pi (v(f)-\hat{v}_r)/v_a}]^2}, \tag{15}$$

In this study, we compared the Doppler estimates before and after aliasing correction using different aliasing correction methods (CS and CP) based on simulation experiments (Sim5). $v_r$ was configured to be 16.8 ($v_a - 10$), 21.8 ($v_a - 5$), 23.8 ($v_a - 3$), and 25.8 ($v_a - 1$) m/s, facilitating the analysis of the impact of aliasing and the corresponding correction performance under different $v_r$. $\sigma_v$ was set to a relatively moderate 2.5 m/s. The setting of the other parameters was similar to that of the other above-mentioned simulation experiments and will not be repeated.

The difference between the Doppler estimates of the simulated I/Q data and input of the simulation ($\Delta v_r$ and $\Delta \sigma_v$) is shown in Figure 8 by means of a violin plot, where the blue, orange, and yellow circles represent the results before correction, after correction using CS, and after correction using CP, respectively. It is worth mentioning that velocity unfolding was conducted on $\Delta v_r$, that is, $\Delta v_r$ was added to $2v_a$ when it was less than $-v_a$. The mean and SD of each distribution shown in Figure 8 are listed in Table 5 to provide a quantitative comparison.

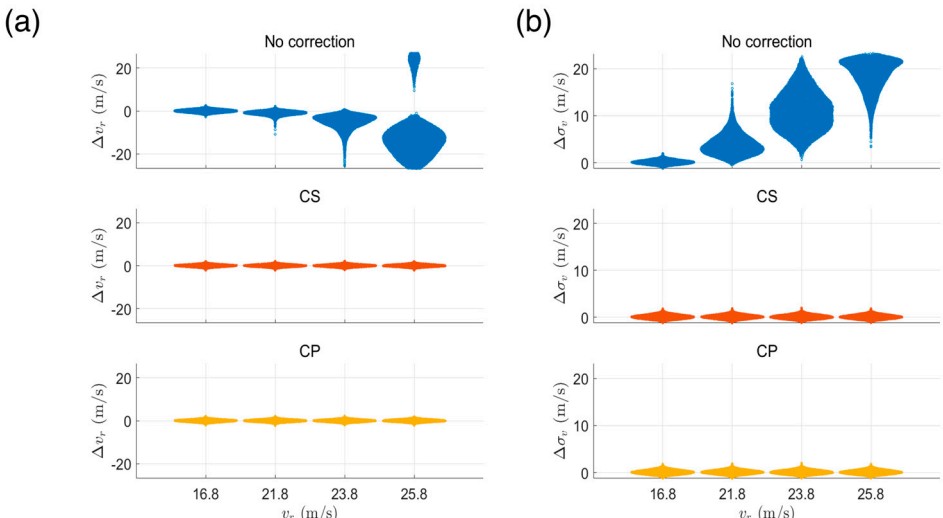

**Figure 8.** Difference between the Doppler estimates of the simulated I/Q data and the input of the simulation for different $v_r$. (**a**) For $\Delta v_r$ and (**b**) For $\Delta \sigma_v$. The blue, orange, and yellow circles denote differences before correction, after correction using CS, and after correction using CP, respectively.

**Table 5.** Mean and SD of the difference between the Doppler estimates of the simulated I/Q data and the input of the simulation for different $v_r$ in Sim5.

| $v_r$ | | **Mean** | | | | **SD** | | | |
|---|---|---|---|---|---|---|---|---|---|
| | | **16.8** | **21.8** | **23.8** | **25.8** | **16.8** | **21.8** | **23.8** | **25.8** |
| No correction | $\Delta v_r$ | −0.025 | −1.018 | −5.263 | −11.285 | 0.708 | 1.029 | 3.448 | 11.906 |
| | $\Delta \sigma_v$ | 0.141 | 3.892 | 11.057 | 18.635 | 0.434 | 2.233 | 4.085 | 3.257 |
| CS | $\Delta v_r$ | 0.01 | 0.002 | 0.012 | 0.005 | 0.551 | 0.555 | 0.55 | 0.549 |
| | $\Delta \sigma_v$ | 0.104 | 0.107 | 0.112 | 0.108 | 0.434 | 0.435 | 0.435 | 0.433 |
| CP | $\Delta v_r$ | 0.006 | −0.003 | 0.008 | 0.001 | 0.561 | 0.565 | 0.56 | 0.558 |
| | $\Delta \sigma_v$ | 0.103 | 0.106 | 0.111 | 0.108 | 0.434 | 0.434 | 0.435 | 0.433 |

As shown in Figure 8 and Table 5, both the Doppler estimates before and after aliasing correction exhibit good and similar estimation performances (i.e., the mean and SD of $\Delta v_r$ and $\Delta \sigma_v$ are around 0 m/s) when $v_r$ is 16.8 m/s because the distribution of the power spectrum is far removed from $v_a$, such that almost no spectrum aliasing exists. As $v_r$ increases to 21.8 m/s, the Doppler estimates before and after the aliasing correction begin to diverge. Specifically, the performance of Doppler estimates after aliasing correction remains the same as those when $v_r$ is 16.8 m/s, while the $v_r$ ($\sigma_v$) estimates before aliasing correction exhibit negative (positive) bias. This can be considered to be a small portion of the aliased power spectrum. When $v_r$ increases to 23.5 and 25.5 m/s, the proportion of power spectrum aliasing increases considerably, resulting in a more evident bias of Doppler estimates, $\Delta \sigma_v$ even exceeding 20 m/s. It should be noted that when $v_r$ is 25.5 m/s, the $v_r$ estimate sometimes has a positive bias. This is because when the power spectrum is split into two parts with similar sizes owing to aliasing and the proportion of the negative velocity side is higher, the $v_r$ estimate is a negative value. Thus, $\Delta v_r$ will be a positive value smaller than $v_a$ after velocity unfolding. In contrast to the situation before aliasing correction, the performance of Doppler estimates after aliasing correction remains unchanged for all $v_r$. In addition, the correction performance of CS and CP are similar, whether based on the qualitative comparison in Figure 8 or the quantitative comparison in Table 5.

In summary, the two proposed methods satisfactorily correct the spectrum aliasing, such that the performance of Doppler estimates of FDP is independent of the value of $v_r$,

which is beneficial to improving the performance of Doppler estimates when $v_r$ is close to the edge of the measurement range. However, considering factors such as algorithm complexity, CP is used for spectrum-aliasing correction in the follow-up to this paper.

### 3.3. Noise Correction

The noise received by the radar can be divided into internal and external sources [13]. Noise power from outside the radar includes thermal radiation from the ground, sun, sky, and precipitation. Noise power within the radar originates from the semiconductor noise, thermal noise of ohmic resistances or conductance, and noise current of the charge carrier currents. Noise affects the estimation of the radar variables; however, the extent of its influence is not constant. The estimation accuracy of the radar variables exhibits good performance even without noise correction when the $SNR$ is sufficiently large (e.g., greater than 20 dB), whereas noise will have a considerable impact on the estimation accuracy of the radar variables when the $SNR$ is low, particularly for polarimetric variables [37]. As mentioned in Section 1, many studies on TDP have focused on alleviating the influence of noise on parameter estimation [10,11], which is sufficient to reflect the importance of noise correction.

As expressed in Equation (9), the noise correction in FDP involves subtracting the noise power per discrete frequency (i.e., $N_{h,v}$ divided by $M$) from $\hat{S}_{h,v}(f)$. However, certain details require further discussion. When the $SNR$ is low, $\hat{S}_{h,v}(f)$ at some discrete frequencies may be lower than the noise power. At this time, the intuitive processing method uses the zero-truncated method (ZT) because negative power has no physical meaning. However, under the assumption that $N_{h,v}$ is accurately measured, this can lead to incomplete noise removal, thus introducing positive bias in the $P_h$ estimates, and further causing bias in some other radar variable estimates (e.g., $Z_{DR}$ and $\rho_{HV}$).

In this study, we proposed a hybrid noise correction method (HY). Specifically, for the estimation of $P_h$, $Z_{DR}$, and $\rho_{HV}$, negative power at some discrete frequencies is allowed during the noise correction process, such that the noise can be completely removed. However, for Doppler estimates, noise correction still uses the ZT method because the calculation involves using $\hat{S}_h(f)$ as a weight. Noise correction is not involved in $\phi_{DP}$ estimation owing to the $\hat{S}_{hv}(f)$ not being affected by noise.

This study compared the estimation performance of FDP before and after noise correction using different noise correction methods (ZT and HY) through simulation experiments (Sim6). $P_h$ was set from 0 to 30 dB in steps of 5 dB to analyze the noise correction performance under different $SNR$ values. The polarimetric variable values were set to the averages of those used in Sim1 and Sim2 to ensure compatibility with both types of precipitation. The setting of the other parameters was similar to that of the other above-mentioned simulation experiments and will not be repeated.

The bias and SD of the estimation of the radar variables are shown in Figures 9 and 10, respectively, where the blue, red, and yellow lines represent the results without noise correction, using the ZT and HY methods, respectively. When the estimation performance of a specific radar variable is the same under different noise processing methods (e.g., the $v_r$ estimates using the ZT and HY methods), only the bias and SD of one method are shown. It is evident from Figure 9 that for all radar variables (except $\phi_{DP}$), the estimated results without noise correction have maximum bias. Additionally, compared with the ZT method, the HY method has better correction performance on estimations of $P_h$, $Z_{DR}$, and $\rho_{HV}$. Notably, when the $SNR$ exceeds 20 dB, the three lines tend to coincide and it is no longer important to determine which noise correction method should be used or whether to perform noise correction.

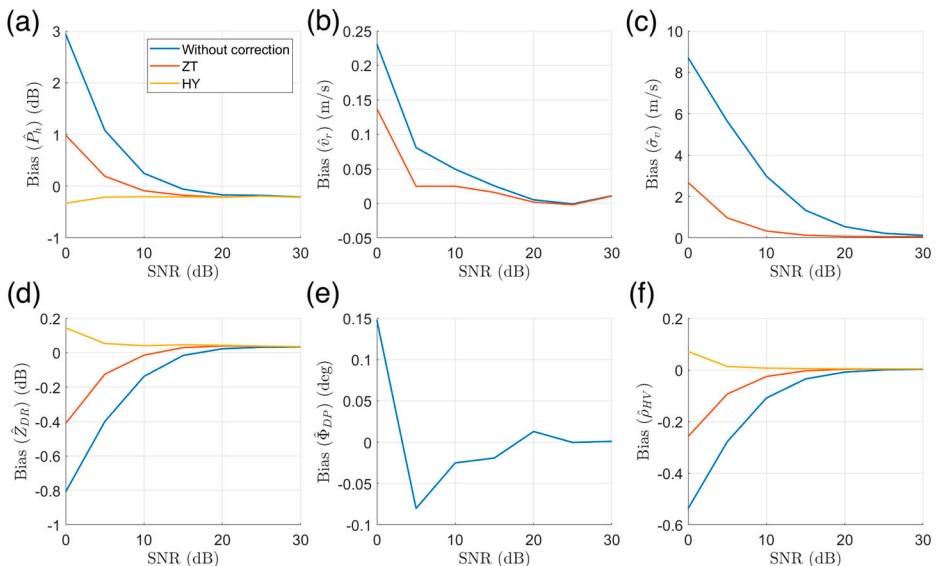

**Figure 9.** Bias of FDP for different *SNR* and noise processing methods. (**a**) $\hat{P}_h$; (**b**) $\hat{v}_r$; (**c**) $\hat{\sigma}_v$; (**d**) $\hat{Z}_{DR}$; (**e**) $\hat{\phi}_{DP}$; and (**f**) $\hat{\rho}_{HV}$.

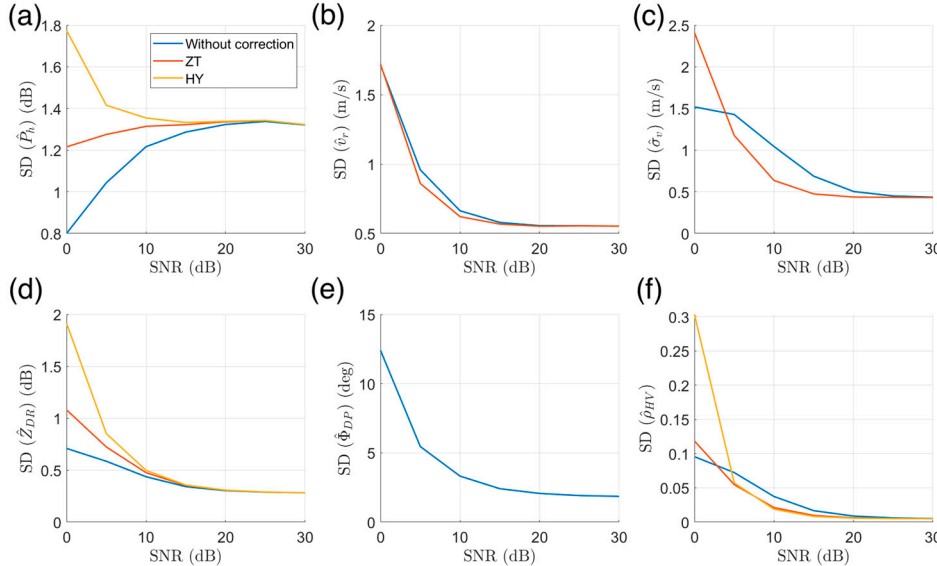

**Figure 10.** SD of FDP for different *SNR* and noise processing methods. (**a**) $\hat{P}_h$; (**b**) $\hat{v}_r$; (**c**) $\hat{\sigma}_v$; (**d**) $\hat{Z}_{DR}$; (**e**) $\hat{\phi}_{DP}$; and (**f**) $\hat{\rho}_{HV}$.

As shown in Figure 10, the SDs of the estimations of $P_h$, $Z_{DR}$, and $\rho_{HV}$ show a different characteristic from their bias under the three different noise processing methods—that is, the HY method has the maximum SD, whereas the ZT method takes second place, and the minimum SD occurs without noise correction.

To understand this phenomenon, we performed a statistical analysis of the $P_h$ estimates under different noise processing methods when the *SNR* was 0 dB, as shown by the histogram in Figure 11. It is evident that the $P_h$ estimates without noise correction have the narrowest distribution, and the distribution is concentrated in the region greater than 0 dB. After noise correction, the distribution shifts to the negative side and gradually widens. Because the noise correction performance of the HY method is superior to that of the ZT method, the $P_h$ estimates obtained using the HY method have more offsets, and the distribution is approximately centered at 0 dB. The 10,000 $P_h$ estimates appear as a distribution rather than as a single value because of the randomness introduced in the I/Q data simulation (Steps 5 and 9 in Section 2.1).

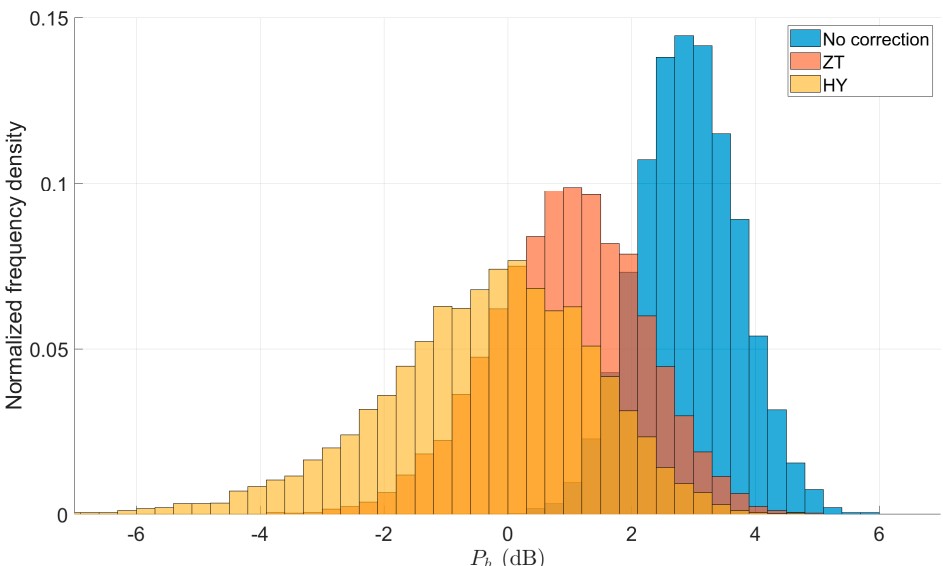

**Figure 11.** $P_h$ estimates under different noise processing methods when the $SNR$ is 0 dB.

When performing noise correction, the expected noise power (input of the simulation) was subtracted from the total power rather than from the actual noise power. This introduced additional randomness and increased the possible value range of $P_h$ estimates, which is characterized by distribution broadening and an increase in the SD. Therefore, the SD of the estimation of $P_h$, $Z_{DR}$, and $\rho_{HV}$ with noise correction can be expected to improve for actual observations. As shown in Figure 10b,c, except for the $\sigma_v$ estimates with an $SNR$ of 0 dB, the SD of the Doppler estimates exhibited little difference before and after noise correction. The increase in the SD of the $\sigma_v$ estimates with noise correction when the $SNR$ was 0 dB was due to the randomness caused by the residual noise. By contrast, $v_r$ estimates were less affected by residual noise.

In summary, the parameter estimation performance improved after noise correction and the HY method exhibited better performance in the $P_h$, $Z_{DR}$, and $\rho_{HV}$ estimates than the ZT method.

## 4. FDP and TDP Performance Comparison

Currently, TDP is widely used in operational weather radars [22]. To prove that FDP has the potential to replace TDP in operational applications, it is necessary to adequately compare the performances of the two to clarify the advantages of FDP over TDP. Moreover, the remaining FDP defects need to be improved. It is worth mentioning that this study only focuses on the difference between FDP and TDP performance (i.e., $v_r$ and $\sigma_v$ estimates), whereas the estimation of other radar variables with the same FDP and TDP performance is not analyzed here, as it has been fully analyzed in existing research [8,38].

### 4.1. Based on Simulated I/Q Data

#### 4.1.1. Gaussian Power Spectrum

First, the weather signals were analyzed under ideal conditions (i.e., the power spectrum had a Gaussian distribution). The performance of FDP and TDP under different $\sigma_v$ (from 0.5 to 4.5 m/s in steps of 0.5 m/s), $SNR$ (from 0 to 30 dB in steps of 5 dB), and $M$ (16, 32, 64, 128, and 256) were compared using three simulation experiments (Sim7, Sim8, and Sim9). As presented in Table 2, only one of the above three parameters was set as a variable in each simulation, with the other two parameters being set to their ideal values. Because only the $v_r$ and $\sigma_v$ estimates were analyzed, the polarimetric variables were not set in the simulations. The FDP parameter settings are listed in Table 3, whereas TDP was performed strictly based on the method described in Section 2.2.

As shown in Figure 12, the Doppler estimates of FDP and TDP exhibit little difference in most cases under different parameter configurations—that is, the difference in the bias or SD between the two is less than 0.5 m/s. The main difference between the two methods is reflected primarily in the SD of $\sigma_v$ estimates at different $\sigma_v$ (Figure 12d), the bias of $\sigma_v$ estimates and SD of $v_r$ estimates at different $SNR$s (Figure 12f,g), the bias in $\sigma_v$ estimates and SD of $v_r$ and $\sigma_v$ estimates at different $M$ (Figure 12j,k,l), respectively. As shown in Figure 12d, the SD of the $\sigma_v$ estimates of FDP increases linearly with increasing $\sigma_v$, while that of TDP first decreases before increasing. When $\sigma_v$ is 0.5 m /s, the SD of $\sigma_v$ estimates of TDP is approximately 0.4 m/s larger than that of FDP, which shows that FDP has an advantage when $\sigma_v$ is low. As shown in Figure 12f,g,j,k, when $SNR$ is low (less than 10 dB) or $M$ is less than 64, FDP has a larger bias in $\sigma_v$ estimates and SD in $v_r$ estimates than those of TDP. However, as shown in Figure 12l, when $M$ is less than 64, the $\sigma_v$ estimate of FDP has a smaller SD than that of TDP.

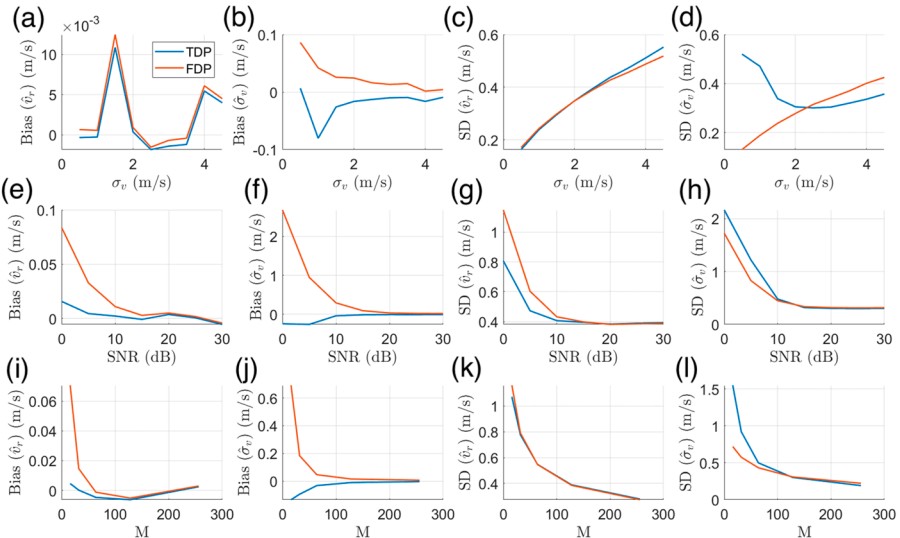

**Figure 12.** Bias and SD of Doppler estimates based on FDP and TDP for different $\sigma_v$, $SNR$, and $M$. (**a**) Bias of $\hat{v}_r$ for different $\sigma_v$; (**b**) bias of $\hat{\sigma}_v$ for different $\sigma_v$; (**c**) SD of $\hat{v}_r$ for different $\sigma_v$; (**d**) SD of $\hat{\sigma}_v$ for different $\sigma_v$; (**e**) bias of $\hat{v}_r$ for different $SNR$; (**f**) bias of $\hat{\sigma}_v$ for different $SNR$; (**g**) SD of $\hat{v}_r$ for different $SNR$; (**h**) SD of $\hat{\sigma}_v$ for different $SNR$; (**i**) bias of $\hat{v}_r$ for different $M$; (**j**) bias of $\hat{\sigma}_v$ for different $M$; (**k**) SD of $\hat{v}_r$ for different $M$; and (**l**) SD of $\hat{\sigma}_v$ for different $M$.

In summary, when $\sigma_v$ is low, FDP is more advantageous than TDP in $\sigma_v$ estimates, but when $SNR$ is low or $M$ is small, the Doppler estimates performance of FDP still has a certain gap compared with that of TDP.

### 4.1.2. Non-Gaussian Power Spectrum

We then compared the performances of FDP and TDP under non-Gaussian power spectrum conditions. Non-Gaussian power spectrum signals are not generated directly by simulation, but by combining several Gaussian power spectrum signals, as has been applied in some existing studies [19,32]. As mentioned in Section 1, one of the advantages of FDP over TDP is that no assumptions are required regarding the distribution of the power spectrum. Consequently, we used the FDP estimation results as a baseline, and the difference between them and the TDP estimation results was considered a bias.

The simulations primarily involved two types of non-Gaussian power spectrum signals—namely, asymmetric power spectrum signals (Sim10) and bimodal power spectrum signals (Sim11). Asymmetric power spectrum signals have been observed in some hailstorms [20], and their distribution appears to broaden and tilt toward one side of the positive or negative velocity. In Sim10, the asymmetric power spectrum signals were generated by combining three Gaussian power spectrum signals, a typical example of which is

shown in Figure 13a. $P_h$ ($v_r$) of the three Gaussian power spectrum signals were set at 30, 25, and 20 dB ($-12$, 0, 12 m/s), respectively, to simulate the tilt characteristics. Bimodal power spectrum signals have appeared in actual observations and numerical simulations of tornados [18,19], their distribution appearing as two Gaussian power spectrum signals with similar amplitudes. In Sim11, we generated bimodal power spectrum signals by combining two Gaussian power spectrum signals, a typical example of which is shown in Figure 13b. $P_h$ of the two Gaussian power spectrum signals was set at 30 dB, and $v_r$ was set at $-10$ and 10 m/s, respectively. The configurations of the other parameters listed in Tables 2 and 3 are similar to those of other simulation experiments and are not explained in detail.

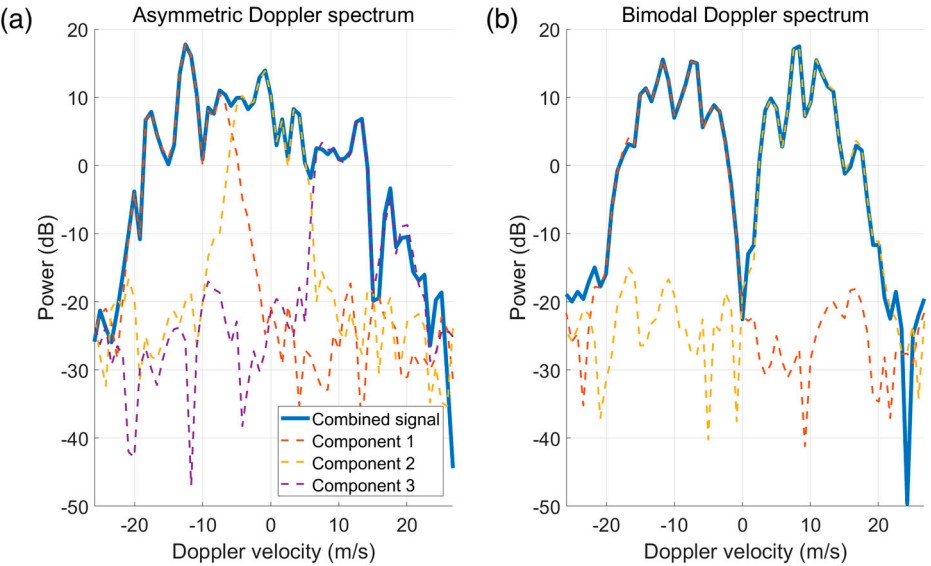

**Figure 13.** Typical non-Gaussian power spectrum signals based on simulations. (**a**) Asymmetric power spectrum signal; (**b**) Bimodal power spectrum signal.

The biases of the Doppler estimates of TDP relative to that of FDP ($\Delta v_r$ and $\Delta \sigma_v$) for the two non-Gaussian power spectrum signals are shown in Figure 14; the mean and SD of $v_r$ and $\sigma_v$ of FDP for an asymmetric (bimodal) power spectrum signal are $-7.379$ and 7.778 m/s (0.015 and 10.094 m/s), respectively. As shown in Figure 14a, for the asymmetric power spectrum signal, the $v_r$ estimates of TDP have a relatively large bias ($-1.476$ m/s), but the distribution of the bias is relatively concentrated (SD is 0.424 m/s). By contrast, the $v_r$ estimates of the TDP for the bimodal power spectrum signals exhibit unbiased characteristics, but the estimation results are not stable, with an SD of up to 1.706 m/s. As shown in Figure 14b, for the asymmetric power spectrum signal, the $\sigma_v$ estimates of TDP exhibit a small negative bias ($-0.38$ m/s), but those of TDP for the bimodal power spectrum signal exhibit a positive bias of up to 1.275 m/s. For both non-Gaussian power spectrum signals, the SD of $\sigma_v$ estimates of TDP relative to that of FDP is approximately 0.7 m/s.

In summary, for non-Gaussian (e.g., asymmetric or bimodal) power spectrum signals, the Doppler estimate results of TDP can be biased or fluctuate considerably if FDP is used as the benchmark, indirectly indicating that for non-Gaussian power spectrum signals, FDP has more advantages over TDP in Doppler estimates.

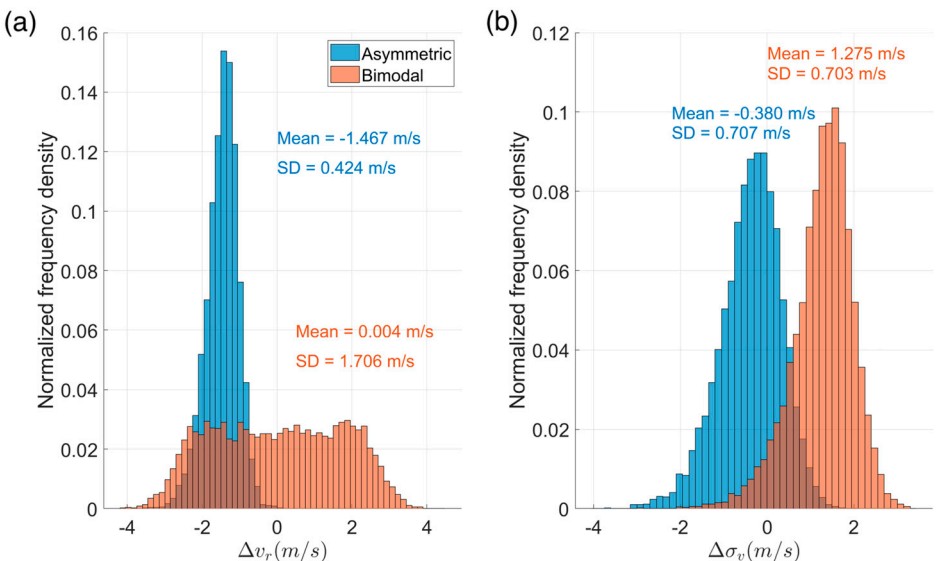

**Figure 14.** Distribution of the Doppler estimates of TDP relative to that of FDP for the two non-Gaussian power spectrum signals. (**a**) $\Delta v_r$; (**b**) $\Delta \sigma_v$.

### 4.2. Based on Measured I/Q Data

In addition to the I/Q data generated based on the simulations, I/Q data observed by the S-band dual-polarization standard weather radar deployed in Changsha (CSSR; operated by the China Meteorological Administration for weather radar calibration applications) were used for the comparative evaluation of FDP and TDP. The CSSR observed a severe storm using the range-height indicator mode at 0728 UTC on 9 August 2023. The estimation of $Z_H$, $v_r$, and $\sigma_v$ based on FDP and the difference in Doppler estimates between FDP and TDP ($\Delta v_r$ and $\Delta \sigma_v$) with an $M$ of 64 are shown in Figure 15. The maximum $Z_H$ exceeds 60 dBZ and $\sigma_v$ is up to 10 m/s above the severe echo (the region highlighted by the red dotted line), which can be regarded as an indicator of the non-Gaussian power spectrum to a certain extent. As shown in Figure 15e,d, $\Delta v_r$ and $\Delta \sigma_v$ are close to 0 m/s in most regions, indicating that the power spectrum at these positions presents an approximate Gaussian distribution. However, as $\sigma_v$ increases, the absolute values of $\Delta v_r$ and $\Delta \sigma_v$ in some regions appear to have abnormally large values, particularly in the regions with large $\sigma_v$ (highlighted by the red dotted line), indicating that the power spectrum deviates from the Gaussian distribution.

We selected two typical range gates from the region highlighted by the red dotted line in Figure 15 and plotted their power spectra in Figure 16. The power spectrum shown in Figure 16a has a distribution similar to that shown in Figure 13a—that is, it is wide and has a certain slope at the top (the red dotted line is an auxiliary line for easy understanding). The $v_r$ and $\sigma_v$ estimates based on FDP are 0.471 and 8.852 m/s, and $\Delta v_r$ and $\Delta \sigma_v$ are $-2.281$ and 0.358 m/s, respectively. The power spectrum shown in Figure 16b has a multi-peak structure (obtained by subjectively adding three semi-ellipses to assist in identifying the positions of the peaks). The $v_r$ and $\sigma_v$ estimates obtained based on FDP are 9.766 and 9.494 m/s, and $\Delta v_r$ and $\Delta \sigma_v$ are 1.128 and 2.631 m/s, respectively.

It is evident from the above analysis that the conclusions based on actual observations are consistent with those of simulations and existing research—that is, the performances of FDP and TDP are almost the same for the Gaussian power spectrum, whereas there is a certain deviation in the estimation results of TDP relative to those of FDP for the non-Gaussian power spectrum, further demonstrating that FDP generates better Doppler estimates than TDP for the non-Gaussian power spectrum.

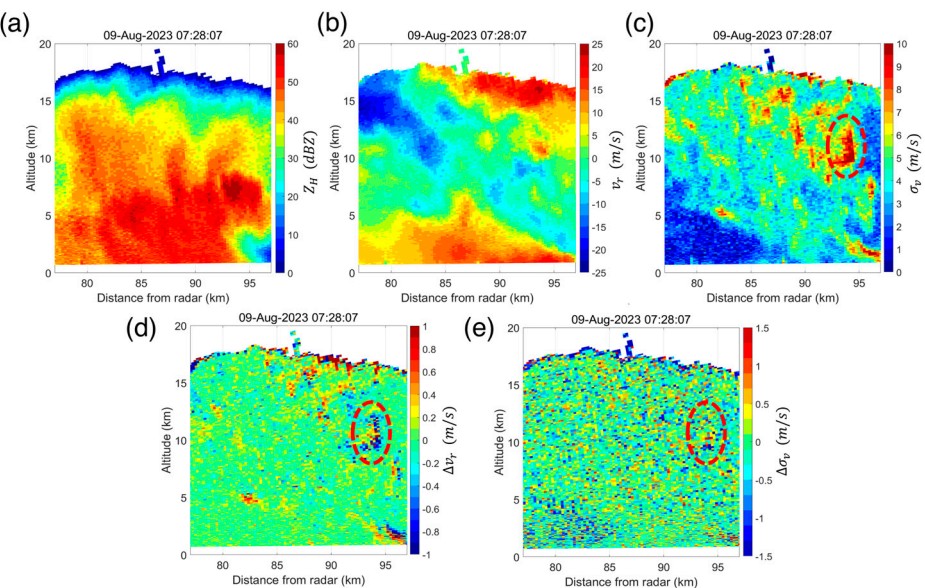

**Figure 15.** A severe storm observed by CSSR using the range-height indicator mode at 0728 UTC on 9 August 2023. (**a**) $Z_H$; (**b**) $v_r$; (**c**) $\sigma_v$; (**d**) $\Delta v_r$; and (**e**) $\Delta \sigma_v$. The red dotted lines highlight the region with large $\sigma_v$.

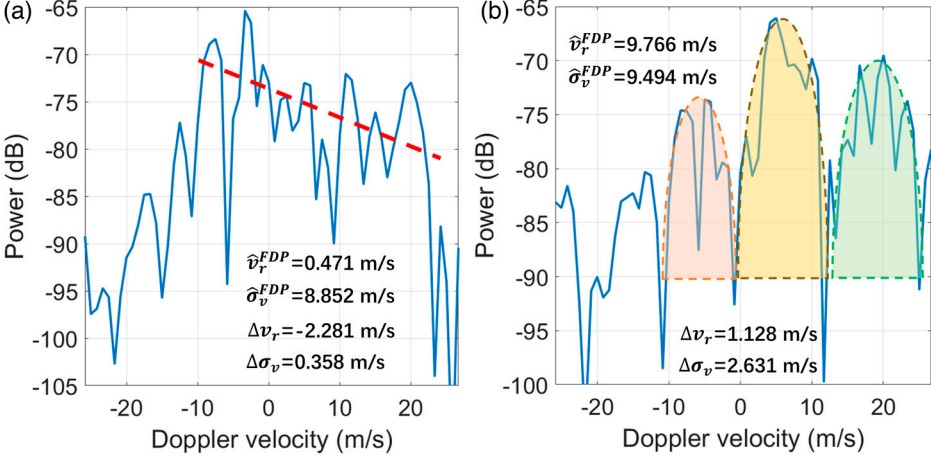

**Figure 16.** Observed non-Gaussian power spectrum signals. (**a**) Asymmetric power spectrum signal; (**b**) multi-peak power spectrum signal. The red dotted line is an auxiliary line for indicating the tilt at the top of the power spectrum. Three semi-ellipses with different colors to assist in identifying the positions of the peaks.

## 5. Discussion

Currently, TDP is widely used for parameter estimation of operational weather radars owing to its high efficiency and robustness. Compared with TDP, the primary advantage of FDP is that it has a seamless connection to spectrum analysis and does not require assumptions about its power spectrum model. However, some specific FDP steps have not been described in detail in existing studies, and it is still unclear whether its performance satisfies the requirements of weather radar operational applications. Consequently, this study focused on these two issues.

This study introduced the technical details of FDP from three perspectives—that is, the window function selection, spectrum-aliasing correction, and noise correction. Additionally, the performance of FDP and the comparison between FDP and TDP were analyzed through 11 simulations and actual observations of the CSSR.

Although the evaluation results presented in this paper showed that FDP has potential for operational applications, problems that must be improved during the research process still exist; for example, when weather and clutter signals are mixed, windowing processing must be performed to reduce the influence of the spectrum leakage of the clutter signal. Thus, how to balance this with the estimation performance needs to be studied further. Moreover, the Doppler estimates performance of FDP still has a certain gap compared with that of TDP under some specific conditions, e.g., $SNR$ is low or $M$ is small. In addition to improving the shortcomings of FDP through the continuous development of new techniques, combining FDP and TDP in a similar way to [39] appears to be a promising short-term compromise.

It is worth mentioning that the study of FDP presented in this paper is only the first step, and the FDP performance will further be improved, and an attempt to use it in an operational environment will be conducted in the future.

## 6. Conclusions

The main results of this study can be summarized as follows:

1. The use of window functions (except for the rectangular window) in performing DFT was beneficial for improving the accuracy of $\sigma_v$ estimates. For other radar variables, the use of window functions (except for the rectangular window) resulted in a decrease in the number of effective samples and an increase in the SD. Therefore, two types of DFT were performed in FDP, one for $\sigma_v$ estimates using a window function with a low taper (the default being the Hamming window), and another for other radar variables estimates using a rectangular window;

2. Both aliasing correction methods described in this paper satisfactorily corrected the spectrum aliasing, such that the performance of Doppler estimates of FDP was independent of the value of $v_r$, which was beneficial in improving the performance of Doppler estimates when $v_r$ was close to the edge of the measurement range. However, owing to advantages such as algorithm complexity, CP should be a better choice for operational applications;

3. The parameter estimation performance improved after noise correction, and the HY method introduced in this study exhibited a better performance in $P_h$, $Z_{DR}$, and $\rho_{HV}$ estimates than the ZT method;

4. For Gaussian power spectrum signals, FDP was more advantageous than TDP in $\sigma_v$ estimates when $\sigma_v$ was low, while the Doppler estimate performance of FDP exhibited a certain gap compared to that of TDP when the $SNR$ was low or $M$ was small;

5. For non-Gaussian (e.g., asymmetric or multi-peak) power spectrum signals, the Doppler estimate results of TDP were biased or fluctuated considerably if FDP was used as the benchmark, indirectly demonstrating that FDP exhibited more advantages than TDP in Doppler estimates for non-Gaussian power spectrum signals.

**Author Contributions:** Conceptualization, S.Z. and Y.C.; methodology and investigation, S.Z. and J.C.; data curation, H.Y. and H.W.; writing—review and editing, Z.S. and L.L. All authors have read and agreed to the published version of the manuscript.

**Funding:** This work was supported by the Youth Science & Technology Project of the Meteorological Observation Centre of the China Meteorological Administration (Grant No. 202304), the Ecological Environment Research Project of Jiangsu Province (Grant No. 2022007), and the Science & Technology Project of Beijing Meteorological Bureau (Grant No. 202204001).

**Data Availability Statement:** The data presented in this study are available on request from the corresponding author. The data are not publicly available due to privacy.

**Conflicts of Interest:** Author Haifeng Yu was employed by the company Huayun Metstar Radar Co., Ltd. The remaining authors declare that the research was conducted in the absence of any commercial or financial relationships that could be construed as a potential conflict of interest.

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
