# Peer review of "Weather Radar Parameter Estimation Based on Frequency Domain Processing: Technical Details and Performance Evaluation"

_remotesensing, doi:10.3390/rs15235624_

Round 1

Reviewer 1 Report

Comments and Suggestions for Authors

The paper is well written. The authors may want to consider repositioning it as a technical note or a communication paper. 

This paper introduces the frequency-domain processing method employed in weather radar signal processing. Especially, it describes several technical details, including window function selection, aliasing correction, and noise correction. The FDP techniques were evaluated with simulated and measured radar data and compared with time-domain processing results.

The paper is very well written and organized. This is a good introductory paper on the Frequency-Domain Processing (FDP) method. However, the FDP has been already introduced in radar community and studied widely. Any new method is NOT suggested in this paper. Therefore, it might be better suited as a technical note or a communication paper.

Minor comments:

1.     Page 2 line 95, lighting or lightning? Please check it.

Reviewer 2 Report

Comments and Suggestions for Authors

This article analyzed the application of window function selection, aliasing correction, and noise correction to frequency-domain processing (FDP) used in weather radar field. Through the analysis of simulation data and the verification of measured data, the article draws a conclusion that FDP has potential for operational environment. In the experiment part, the authors use the control variable method, with two indicators, bias and SD, to measure the error of the estimation of different parameters in convective precipitation and stratiform precipitation. In addition, the effects of aliasing correction and noise correction on the estimation of relevant parameters are analyzed. For Gaussian and non-Gaussian power spectrum signals, it is verified that FDP is better than TDP in some cases.

Q1: Could the authors compare the spectrum aliasing correction method with other methods?

Q2: In the noise correction part, for different parameter estimates, can the same signal be denoised in different ways to obtain as many accurate signal parameter estimates as possible?

Q3: There is only one group of measured data for verification. To enhance the persuasion of the conclusion of the paper, I think more data is needed in the experimental analysis.

Q4: It is suggested to add more references of recent years.

Comments on the Quality of English Language

none

Reviewer 3 Report

Comments and Suggestions for Authors

The paper "Weather radar parameter estimation based on frequency domain processing: technical details and performance evaluation" by Zhang et al. is fascinating, and I congratulate the authors for their work.

The paper uses several technical details in frequency-domain processing including window function selection, aliasing correction, and noise correction.

Please find here some comments on the paper:

An illustrative figure would be very welcome in the Introduction.

Table 2 needs to be more straightforward. The authors must reorganize it.

Figure 5 - please keep the figures as clean as possible - the details on Mean and SD should be presented in the label.

Line 313: The difference between the Doppler estimates of the simulated I/Q data and the input of the simulation is shown in Figure 7 by means of a violin plot. Authors must present a quantitative comparison, for example, using a hypothesis test such as the Mann–Whitney U test.

The paper needs a better discussion of the results: is the literature compatible with the findings and appointments?

What are the limitations of the research? Moreover, what are the perspectives?

Please highlight the paper's innovation when compared to the previous contributions.

My last question is about the different technicals (window function selection, aliasing correction, and noise correction): are the technicals completely independent? What is the combined effect of them?

Comments on the Quality of English Language

Minor editing of English language required

Round 2

Reviewer 3 Report

Comments and Suggestions for Authors

Accept in present form

Comments on the Quality of English Language

Minor editing of English language required